# THE MAP EQUATION GOES NEURAL

## ABSTRACT

Community detection and graph clustering are essential for unsupervised data exploration and understanding the high-level organisation of networked systems. Recently, graph clustering has received attention as a primary task for graph neural networks. Although hierarchical graph pooling has been shown to improve performance in graph and node classification tasks, it performs poorly in identifying meaningful clusters. Community detection has a long history in network science, but typically relies on optimising objective functions with custom-tailored search algorithms, not leveraging recent advances in deep learning, particularly from graph neural networks. In this paper, we narrow this gap between the deep learning and network science communities. We consider the map equation, an information-theoretic objective function for unsupervised community detection. Expressing it in a fully differentiable tensor form that produces soft cluster assignments, we optimise the map equation with deep learning through gradient descent. More specifically, the reformulated map equation is a loss function compatible with any graph neural network architecture, enabling flexible clustering and graph pooling that clusters both graph structure and data features in an end-to-end way, automatically finding an optimum number of clusters without explicit regularisation by following the minimum description length principle. We evaluate our approach experimentally using different neural network architectures for unsupervised clustering in synthetic and real data. Our results show that our approach achieves competitive performance against baselines, naturally detects overlapping communities, and avoids over-partitioning sparse graphs.

## 1 INTRODUCTION

Graph neural networks have enabled applying deep learning to graph-structured data by incorporating elements of the graph structure in the computational graph of the neural network (Scarselli et al., 2009; Gilmer et al., 2017; Kipf & Welling, 2017). An important application of learning methods for graph-structured data is graph clustering, which involves grouping the set of nodes in a graph into sets, or *clusters*, of "related" nodes (Schaeffer, 2007). A closely related problem in network science is studying graph structure at the mesoscale (Griebenow et al., 2019; Petrović et al., 2022; Villegas et al., 2023), in which a key inference problem is detecting *communities* of "similar" nodes. While precise definitions of communities and clusters remain an active research field (Fortunato, 2010; Peixoto & Kirkley, 2022), graph clustering and community detection methods are essential in unsupervised data exploration and in improving performance and scaling in graph neural networks (Chiang et al., 2019; Tsitsulin et al., 2023; Bianchi & Lachi, 2023).

Typical network science approaches use search algorithms to minimise an objective function that characterises what constitutes "good communities". In contrast, embedding-based approaches, such as node2vec (Grover & Leskovec, 2016) and DeepWalk (Perozzi et al., 2014), detect communities by first computing embedding vectors for nodes, followed by clustering in the embedding space, for example, using k-means (Kojaku et al., 2023; Tsitsulin et al., 2023). Recently, the community detection approach known as modularity (Newman, 2006) has been integrated with graph neural networks, improving deep graph clustering, community-aware node labelling, and link prediction (Murata & Afzal, 2018; Tsitsulin et al., 2023).

Here, we consider the map equation, an information-theoretic objective function for community detection. By expressing the map equation in differentiable tensor form using soft cluster assignments, we enable end-to-end optimisation of the map equation as a loss function with gradient descent. We

evaluate our approach using different graph neural network (GNN) architectures against Infomap, the map equation's stochastic optimisation algorithm, and various (deep) graph clustering methods in synthetic and real networks. We find that with soft cluster assignments, the map equation naturally produces overlapping communities, a task that is typically computationally expensive and requires, for example, higher-order data (Rosvall et al., 2014; Holmgren et al., 2023) or clustering nodes indirectly through clustering links (Evans & Lambiotte, 2009; Ahn et al., 2010). Using gradient descent, we avoid getting stuck in spurious optima in degenerate solution landscapes, a problem that Infomap addresses with a Bayesian approach (Smiljanić et al., 2021).

Our key contributions are:

1. We propose Neuromap, an alternative to the widely-used Infomap algorithm for minimising the map equation in unsupervised clustering tasks, leveraging recent developments in deep learning and GNNs.
2. Our differentiable adaptation of the map equation is a loss function compatible with any GNN architecture to naturally detect overlapping communities, or soft clusters. Adapting the map equation as an information theoretic-based loss function with a clear underlying model allows interpretable end-to-end deep graph clustering that leverages node and edge features to improve real-world performance without requiring explicit regularisation.
3. We evaluate our approach empirically in a suite of synthetic and real-world benchmarks, and show the improvements it proffers over Infomap and other (deep) graph clustering methods, and its potential as an exciting avenue for future work.

## 2 RELATED WORK

**Community Detection and the Map Equation.** The map equation (Rosvall & Bergstrom, 2008) is an objective function for community detection and builds on the minimum description length principle from information theory (Rissanen, 1978). It detects communities by compressing the modular description of a network and has been shown to work well in synthetic and real networks from across domains (Lancichinetti & Fortunato, 2009; Aldecoa & Marín, 2013; Šubelj et al., 2016). The map equation framework has been extended to detect overlapping communities based on observed or modelled higher-order data (Rosvall et al., 2014; Holmgren et al., 2023), avoid over-partitioning in sparse networks using a Bayesian approach (Smiljanić et al., 2021), and to deal with sparse constrained structures (Edler et al., 2022). Moreover, the map equation framework can incorporate node features through an extension (Emmons & Mucha, 2019) or by preprocessing data (Bassolas et al., 2022). Detecting communities relies on Infomap, a stochastic search algorithm that optimises the map equation (Edler et al., 2017). Other popular approaches to community detection include the stochastic block model (SBM) and its variants (Peixoto, 2014), used as a benchmark in our real-world experiments (Tsitsulin et al., 2023), and modularity maximisation which identifies communities by comparing link densities between and within groups of nodes against expected densities according to a null model (Newman, 2006). For a detailed overview of community detection in complex networks, we refer to Fortunato (2010) and Fortunato & Newman (2022).

**Graph Neural Networks and Hierarchical Pooling.** Pooling is an essential component in graph neural networks for tackling graph-level tasks that combine, or coarse-grain, node and edge-level features into a single graph-level vector. Pooling also enables building deeper graph neural networks that perform better on node and graph classification tasks (Ying et al., 2018; Bianchi & Lachi, 2023). Inspired by pooling in convolutional neural networks, hierarchical pooling involves multistep coarse-graining that "pools" groups of nodes into single nodes to obtain coarse-grained representations of a graph. When these groups are meaningful clusters, they have been shown to improve the performance in hierarchical pooling architectures (Bianchi & Lachi, 2023). Consequently, graph pooling has become a focus in GNN research, emphasising the importance of graph clustering as a primary objective for GNNs (Tsitsulin et al., 2023). Chiang et al. (2019) have also shown how graph clustering enables graph neural networks to scale to larger-than-memory graph-structured data by partitioning the graph into subgraphs for splitting into mini-batches and avoiding the full space complexity of training GNNs in full-batch.

**Graph Clustering and (Deep) Representation Learning.** When clustering a graph's nodes into sets of related nodes, those resulting sets may be disjoint, known as "hard" clusters, or they may

overlap, known as "soft" clusters (Ferraro & Giordani, 2020), synonymous with overlapping communities. Graph clustering has interested the machine learning community for at least 2 decades (Yu et al., 2005) and we refer readers to Schaeffer (2007) for a detailed survey of earlier works. One of the earliest approaches involves clustering in the eigenspace of a graph's Laplacian matrix, popularised in the machine learning community by Shi & Malik (2000). Since then, several methods for clustering involving representation learning have been developed (Tandon et al., 2021), where neural representation learning methods such as DeepWalk (Perozzi et al., 2014) and node2vec (Grover & Leskovec, 2016) have proffered better performance in detecting network communities (Kojaku et al., 2023). A more recent development in neural graph clustering is deep graph clustering (Tsitsulin et al., 2023), where clustering objectives are incorporated into specifically designed loss functions used for graph pooling and graph clustering. This approach enables incorporating both graph structure *and* graph features, such as node and edge features, in fully end-to-end optimisation of clustering objectives. Other approaches include graph autoencoders (Wang et al., 2017; Mrabah et al., 2022), contrastive learning (Ahmadi et al., 2022), and self-expressiveness (Bandyopadhyay & Peter, 2021). We refer readers to Yue et al. (2022) and Xing et al. (2022) for recent surveys of deep graph clustering and community detection with deep learning.

## 3 THE MAP EQUATION GOES NEURAL

**The Map Equation.** The map equation (Rosvall & Bergstrom, 2008) is an information-theoretic objective function for unsupervised community detection based on the minimum description length principle (Rissanen, 1978). It formulates community detection as a compression problem and uses random walks as a proxy to model dynamic processes on networks, also called *flow*. The goal is to describe the random walk as efficiently as possible by minimising its expected per-step description length, or *codelength*, through partitioning the network into groups of nodes, called modules, where the random walker tends to stay for a relatively long time. In practice, however, the map equation does not simulate random walks; instead, the codelength is calculated analytically.

Let $G = (V, E, \delta)$ be a graph with nodes $V$, links $E$, and $\delta \colon E \to \mathbb{R}_0^+$ a function that specifies non-negative link weights. When all nodes are assigned to the same module, the codelength, that is the expected minimum number of bits required to encode the random walker's position, is the Shannon entropy $H$ over the nodes' visit rates (Shannon, 1948), $H(P) = -\sum_{u \in V} p_u \log_2 p_u$, where $p_u$ is node $u$'s visit rate and $P$ is the set of node visit rates. In undirected graphs, we compute visit rates directly as $p_u = s_u / \sum_{v \in V} s_v$, where $s_u = \sum_{v \in V} \delta(u, v)$ is node $u$'s strength. In directed graphs, we use smart teleportation (Lambiotte & Rosvall, 2012) and a power iteration to compute node visit rates numerically.

When we reflect the graph's modular structure in how we partition the nodes, the codelength becomes a weighted average of module-level entropies. However, we also need to consider the so-called index level and its entropy for switching between modules. Minimising the map equation means balancing between partitioning the network into many small modules for low module-level entropy and into few large modules for low index-level entropy. This interplay between module- and index-level codelengths naturally implements Occam's razor and prevents trivial solutions where all nodes are assigned to the same module or each node is assigned to a singleton module (Rissanen, 1978). The map equation calculates the codelength for a partition $\mathsf{M}$ as $L(\mathsf{M}) = qH(Q) + \sum_{\mathsf{m} \in \mathsf{M}} p_\mathsf{m} H(P_\mathsf{m})$. Here, $q = \sum_\mathsf{m} q_\mathsf{m}$ is the rate at which the random walker enters modules, $q_\mathsf{m}$ is module $\mathsf{m}$'s entry rate, and $Q = \{q_\mathsf{m} \mid \mathsf{m} \in \mathsf{M}\}$ is the set of module entry rates; $p_\mathsf{m}$ is the rate at which the random walker moves in module $\mathsf{m}$, including the module exit rate $\mathsf{m}_{\text{exit}}$, and $P_\mathsf{m} = \{\mathsf{m}_{\text{exit}}\} \cup \{p_u \mid u \in \mathsf{m}\}$ is the set of module-normalised node visit and exit rates for module $\mathsf{m}$. We provide more details and an example in Appendix A. For our implementation, we use the following way of rewriting the map equation (Rosvall & Bergstrom, 2008),

$$
\begin{aligned}
L(\mathsf{M}) = &\Big(\sum_{\mathsf{m} \in \mathsf{M}} q_\mathsf{m}\Big) \log_2\Big(\sum_{\mathsf{m} \in \mathsf{M}} q_\mathsf{m}\Big) - 2\sum_{\mathsf{m} \in \mathsf{M}} q_\mathsf{m} \log_2 q_\mathsf{m} \\
&- \sum_{u \in V} p_u \log_2 p_u + \sum_{\mathsf{m} \in \mathsf{M}} \Big(\mathsf{m}_{\text{exit}} + \sum_{u \in \mathsf{m}} p_u\Big) \log_2\Big(\mathsf{m}_{\text{exit}} + \sum_{u \in \mathsf{m}} p_u\Big).
\end{aligned} \tag{1}
$$

Inspired by recent work that highlighted the importance of deep graph clustering as a primary graph learning task and introduced modularity as a clustering and pooling objective for graph neural net-

works (Tsitsulin et al., 2023), we express the two-level map equation (Equation (1)) in differentiable tensor form for optimisation with GNNs and gradient descent.

**Notation.** We use the following notation for the remainder of the paper. Boldface italic symbols represent tensors, and non-boldface italic symbols represent scalars. Lowercase italic letter subscripts of tensors such as $i$ and $j$ in $A_{ij}$ represent tensor indices, following Einstein notation (Carroll, 2019). The number of tensor indices of a tensor indicates the rank of a tensor. Lowercase italic letter subscripts of scalars such as $i$ and $j$ in $A_{ij}$ refer to indices of specific elements in tensors. Integer, Roman letter, and uppercase italic letter subscripts are used to distinguish between different tensors, scalars, and functions, e.g. $(\boldsymbol{A}_1)_{ij} = \boldsymbol{A}_1 \neq \boldsymbol{A}_2 = (\boldsymbol{A}_2)_{ij}$. Replacing or fully omitting tensor indices does not change the definition of a tensor, i.e. $\boldsymbol{p}_i$, $\boldsymbol{p}_k$, and $\boldsymbol{p}$ are equivalent. Swapping tensor indices represents a transpose between two dimensions of the tensor, e.g. $\boldsymbol{A}_{ij}{}^T = \boldsymbol{A}_{ji}$. We omit the use of superscript tensor indices and the distinction between covariant and contravariant vectors. Summations over a tensor index, e.g. $\boldsymbol{d}_i = \sum_j \boldsymbol{A}_{ij}$ represents a sum over a tensor dimension, and thereby a reduction of a free tensor index ($j$ in this example). Matrix or tensor multiplications written with tensor indices are written as tensor contractions, e.g. $\boldsymbol{T}_{ij} = \sum_k \boldsymbol{D}_{ik}^{-1}\boldsymbol{A}_{kj}$, summing over a shared tensor index ($k$ in this example). Outer products such as $\boldsymbol{D}_{ik} = \boldsymbol{\delta}_{ik}\boldsymbol{d}_i$ are written without a contraction or sum over a tensor index.

**Map Equation Loss.** Let $\boldsymbol{A}_{ij}$ be the graph's weighted adjacency matrix where each element $A_{ij}$ in the matrix represents the weight on the link from node $i$ to $j$, where $1 \leq i \leq n$ and $1 \leq j \leq n$ are integer node indices, and $n = |V|$ the number of nodes in the graph, and $\boldsymbol{T}_{ij}$ the graph's transition matrix. In undirected graphs, we calculate $\boldsymbol{T}_{ij} = \sum_k \boldsymbol{D}_{ik}^{-1}\boldsymbol{A}_{kj}$, where $\boldsymbol{D}_{ik} = \boldsymbol{\delta}_{ik}\sum_j \boldsymbol{A}_{ij}$ is the diagonal degree matrix, $\boldsymbol{D}_{ik}^{-1}$ is its matrix inverse, and $\boldsymbol{\delta}_{jk}$ is the Kronecker delta whose elements $\delta_{jk} = 1$ where $j = k$ and $\delta_{jk} = 0$ otherwise. The graph' flow matrix $\boldsymbol{F}_{ij} = \sum_k \boldsymbol{T}_{ik}\,\boldsymbol{p}_k\,\boldsymbol{\delta}_{jk}$, which specifies the fraction of flow on each link, corresponding to the probability that a random walker uses each respective link, can be calculated from the transition matrix $\boldsymbol{T}_{ij}$ and $\boldsymbol{p}_i = \frac{1}{d}\sum_j \boldsymbol{A}_{ij}$, which is the vector of node visit rates, and where $d = \sum_{i,j} \boldsymbol{D}_{ij}$ is the total weighted degree of the network. In directed graphs, we use recorded smart teleportation (Lambiotte & Rosvall, 2012), and calculate $\boldsymbol{T}$ as

$$\boldsymbol{T} = \alpha\,\frac{1}{d}\,\boldsymbol{D}\,\mathbf{1}_{n \times n} + (1-\alpha)\,\boldsymbol{T}_{ij} = \alpha\,\frac{1}{d}\,\boldsymbol{D}_{ij}\,\boldsymbol{\delta}_{ij}\,\mathbf{1}_j + (1-\alpha)\sum_k \boldsymbol{D}_{ik}^{-1}\boldsymbol{A}_{kj}, \qquad (2)$$

where $\mathbf{1}_j$ is a vector of ones and $\alpha$ is a teleportation parameter, typically $\alpha = 0.15$. To obtain the flow matrix $\boldsymbol{F}_{ij}$, we calculate $\boldsymbol{p}_i$ with a power iteration from $\boldsymbol{T}_{ij}$.

Let $\boldsymbol{S}_{ia}$ be a soft cluster assignment matrix whose elements $S_{ia}$ encode what fraction of node $i$ is assigned to cluster $a$, $1 \leq i \leq n$ and $1 \leq a \leq s$, where $s$ a specified maximum possible number of clusters. $\boldsymbol{E}_{ab} = \sum_{i,j} \boldsymbol{S}_{ia}^T\,\boldsymbol{F}_{ij}\,\boldsymbol{S}_{jb} = \sum_{i,j} \boldsymbol{S}_{ai}\,\boldsymbol{F}_{ij}\,\boldsymbol{S}_{jb}$ encodes the fraction of flow between each pair of clusters, where $E_{ab}$ is the total flow from cluster $a$ to cluster $b$. Following Equation (1), we define

$$e_1 = \sum_{a,b} \boldsymbol{E}_{ab} - \sum_{a,b} \boldsymbol{\delta}_{ab}\,\boldsymbol{E}_{ab}, \qquad \left(\boldsymbol{e}_2\right)_a = \sum_b \left(\boldsymbol{E}_{ab} - \boldsymbol{\delta}_{ab}\,\boldsymbol{E}_{ab}\right),$$

$$\left(\boldsymbol{e}_3\right)_i = \boldsymbol{p}_i, \qquad \left(\boldsymbol{e}_4\right)_b = \sum_a \left(\boldsymbol{E}_{ab} - \boldsymbol{\delta}_{ab}\,\boldsymbol{E}_{ab}\right) + \sum_a \boldsymbol{E}_{ab}.$$

Because $\boldsymbol{e}_3$ is constant and does not depend on $\boldsymbol{S}$, it can be omitted. Finally, we obtain the map equation in tensor form, where logarithms are applied component-wise,

$$L\left(\boldsymbol{S}, \boldsymbol{F}\right) = e_1 \log_2 e_1 - 2\sum_{a,b}\left(\left(\boldsymbol{e}_2\right)_a \boldsymbol{\delta}_{ab}\,\log_2\left(\boldsymbol{e}_2\right)_b\right)$$
$$- \sum_{i,j}\left(\left(\boldsymbol{e}_3\right)_i \boldsymbol{\delta}_{ij}\,\log_2\left(\boldsymbol{e}_3\right)_j\right) + \sum_{a,b}\left(\left(\boldsymbol{e}_4\right)_a \boldsymbol{\delta}_{ab}\,\log_2\left(\boldsymbol{e}_4\right)_b\right). \qquad (3)$$

**Learning.** We focus on the task of unsupervised graph clustering and set out to learn the cluster assignment matrix $\boldsymbol{S}$ that minimises Equation (3). The optimal number of clusters is chosen

automatically during learning by minimising the map equation. In principle, any neural network architecture, such as a multi-layer perceptron (MLP) or a graph neural network (GNN), can be used to learn $\boldsymbol{S}$. However, the quality of detected clusters depends on the expressivity of the (graph) neural network architecture and node features $\boldsymbol{X}$ used. In the absence of real node features, the identity matrix $\boldsymbol{X}_{ij} = \boldsymbol{\delta}_{ij}$, where $1 \leq i \leq n$ and $1 \leq j \leq n$, can be used as node features, however, designing expressive lower-dimensional node features remains an active research area (Lim et al., 2022). Because the map equation involves logarithms, we need to take care not to have zero values in $\boldsymbol{S}$, which we achieve by adding a small constant $\epsilon$ to each value of the network's output to ensure differentiability.

Unlike other methods that typically require a regularisation term to avoid over-partitioning (Tsitsulin et al., 2023), the map equation naturally incorporates Occam's razor by following the minimum description length principle for balancing between model complexity and fit (Rissanen, 1978).

We note that this approach can be easily adapted for graph pooling (Tsitsulin et al., 2023).

**Complexity**  Similar to DMoN (Tsitsulin et al., 2023), the most expensive calculation in the map equation loss is the pooling operation $\boldsymbol{E}_{ab} = \sum_{i,j} \boldsymbol{S}_{ia}^T \boldsymbol{F}_{ij} \boldsymbol{S}_{jb}$. For undirected networks the complexity is dictated by the sparsity of $\boldsymbol{A}$, that is, the sparsity of the network. When $s \ll n$, and we have a sparse network with $|E| \sim |V|$, the complexity of Neuromap becomes linear in the number of nodes, $\mathcal{O}(n)$. When the network is dense, $|E| \sim |V|^2$, or the maximum number of clusters approaches the number of nodes $s \sim n$, we approach quadratic complexity, $\mathcal{O}(n^2)$. For scalability, we therefore recommend keeping $s \ll n$. For directed networks, $\boldsymbol{F}$ is calculated using recorded smart teleportation (Equation (2)), and is, therefore, a dense square matrix such that the complexity of the map equation loss becomes quadratic, $\mathcal{O}(n^2)$.

## 4  RESULTS

We evaluate our approach on a set of synthetic and real-world unsupervised graph clustering benchmarks, using different neural network architectures to minimise the map equation. We use the Python deep learning and geometric deep learning frameworks PyTorch (Paszke et al., 2019) and PyTorch Geometric Fey & Lenssen (2019), respectively, to implement our models.

### 4.1  SYNTHETIC NETWORKS WITH HARD CLUSTERS

We generate both undirected and directed Lancichinetti-Fortunato-Radicchi (LFR) benchmark networks with planted ground-truth clusters (Lancichinetti et al., 2008) with $N \in \{100, 1000\}$ nodes, average degree $k \in \{\ln N, 2 \ln N\}$, rounded to the nearest integer, maximum degree $k_{\max} = 2\sqrt{N}$, also rounded to the nearest integer, and mixing parameter $\mu$ between $0.05$ and $0.85$ with a step size of $0.05$. We set the power-law exponents for the node degree distribution to $\tau_1 = 2$, and for the community size distribution to $\tau_2 = 1$. For each combination of parameters, we generate 25 LFR networks using the implementation[1] provided by the authors.

We use Infomap and four different neural network architectures to minimise the map equation and detect communities. The first model is an MLP with two hidden layers:

$$\boldsymbol{X}_{\text{out}} = \text{softmax}\left(\text{ReLU}\left(\boldsymbol{X}_{\text{in}}\boldsymbol{W}_1\right)\boldsymbol{W}_2\right). \tag{4}$$

The second model is a graph isomorphism network-inspired GNN (Xu et al., 2019), which we call GIN, with four hidden layers, with message passing in the second and last layers:

$$\boldsymbol{X}_{\text{hidden}} = \text{MLP}\left(\boldsymbol{A}\,\text{ReLU}\left(\boldsymbol{X}_{\text{in}}\boldsymbol{W}_{\text{encoder}}\right)\right), \tag{5}$$

$$\boldsymbol{X}_{\text{out}} = \text{softmax}\left(\boldsymbol{A}\boldsymbol{X}_{\text{hidden}}\boldsymbol{W}_{\text{decoder}}\right), \tag{6}$$

where

$$\text{MLP}\big(\boldsymbol{X}\big) = \text{ReLU}\left(\text{ReLU}\left(\boldsymbol{X}\boldsymbol{W}_1\right)\boldsymbol{W}_2\right).$$

---

[1] https://sites.google.com/site/andrealancichinetti/benchmarks

The third and fourth models, which we call $GNN_1$ and $GNN_2$, respectively, are graph neural network architectures with a single message passing layer and skip-connections between MLP layers:

$$\boldsymbol{X}_{T+1} = \mathrm{MLP}_T\left(\boldsymbol{X}_T\right) + \boldsymbol{A}\boldsymbol{X}_{\mathrm{in}}, \tag{7}$$

$$\boldsymbol{X}_{\mathrm{out}} = \mathrm{softmax}\Big(\mathrm{ReLU}\left(\boldsymbol{X}_T\,\boldsymbol{W}_{T,1}\right)\boldsymbol{W}_{T,2} + \sum_{T=1}^{T_{\mathrm{max}}} \boldsymbol{X}_T\Big), \tag{8}$$

where

$$\mathrm{MLP}_T\left(\boldsymbol{X}_T\right) = \mathrm{ReLU}\left(\mathrm{ReLU}\left(\boldsymbol{X}_T\,\boldsymbol{W}_{T,1}\right)\boldsymbol{W}_{T,2}\right),$$

and $\boldsymbol{X}_{\mathrm{in}} = \boldsymbol{X}_0$. We set the number of layers $T_{\mathrm{max}} = 1$ and $T_{\mathrm{max}} = 2$ for $GNN_1$ and $GNN_2$, respectively.

For each LFR network, we run Infomap and each neural network 10 times, choose the partition with the lowest codelength, and measure their performance in terms of normalised mutual information (NMI) against planted ground truth clusters. We train all models for 1000 epochs with dropout probability 0.1 after each layer, and use a learning rate of 0.0001 for the MLP and GIN, and a learning rate of 0.001 for $GNN_1$ and $GNN_2$. As initial node features, we use the identity matrix, and, for all models, we obtain the soft cluster assignment matrix as $\boldsymbol{S} = \boldsymbol{X}_{\mathrm{out}} + \epsilon$ with $\epsilon = 10^{-8}$.

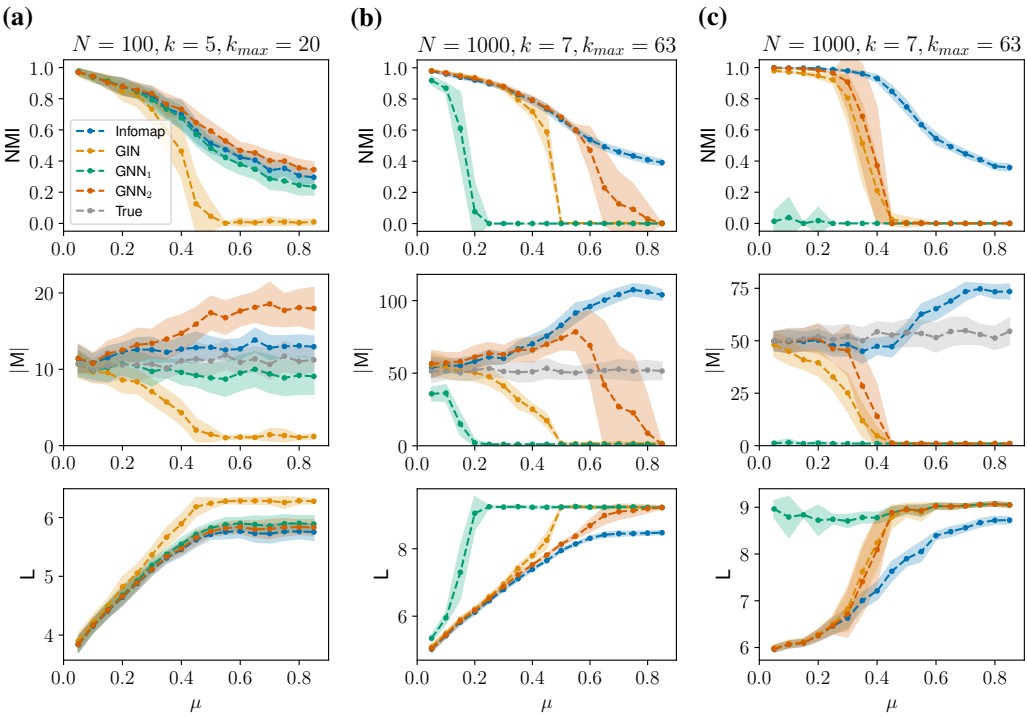

Figure 1: Results for Infomap, GIN, $GNN_1$, and $GNN_2$ on synthetic networks with planted community structure. Column (a) shows results for undirected networks with 100 nodes, (b) for undirected networks with 1000 nodes, and (c) for directed networks with 1000 nodes. Results are averages for partition quality in terms of NMI, number of detected communities $|M|$, and codelength $L$. Error bands show one standard deviation from the mean. We include additional results in Appendix B.

We find that cluster quality depends on the neural network architecture used (Figure 1). In small undirected networks, $GNN_2$ performs best in terms of NMI, followed by Infomap, $GNN_1$, and then GIN; in small directed networks, Infomap performs best, followed by $GNN_2$, $GNN_1$, and then GIN (Appendix B). In larger undirected networks, $GNN_2$ and Infomap perform similarly well for low to medium mixing values. For large mixing values, Infomap generally achieves higher NMI values than the neural network-based approaches; GIN shows the weakest performance, being unable to detect communities for low mixing values in larger networks. In directed networks, Infomap achieves higher NMI values than the neural network-based models. Since the MLP detects a single community in nearly all instances, we omit it from the results. Infomap consistently achieves

the shortest codelength but tends to increasingly over-partition the networks with larger mixing values. Likewise, $GNN_2$ over-partitions the networks, reporting increasingly more communities than in the ground truth for higher $\mu$. GIN and $GNN_1$ detect a single community when $\mu$ becomes large enough, indicating that a regularisation mechanism which avoids over-partitioning is at work. Generally speaking, we find that larger networks require more powerful neural network architectures to achieve good performance in terms of NMI. Similarly, more powerful GNNs perform better for larger values of $\mu$. These observations suggest how practitioners may choose a neural network architecture that fits their research question's needs: simpler GNNs partition the networks into fewer communities, achieving lower NMI values, while more powerful GNNs perform better in terms of NMI, but at the expense of potentially over-partitioning the network.

## 4.2 SYNTHETIC NETWORKS WITH OVERLAPPING AND CONSTRAINED STRUCTURE

We applied our approach to small synthetic networks with overlapping communities. Through soft cluster assignment, we naturally discover overlapping communities (Figure 2a-b). In contrast, the standard map equation with hard cluster assignments requires higher-order data (Rosvall et al., 2014) or flow modelling (Holmgren et al., 2023) and a memory network representation to detect overlapping communities. We include further examples in Appendix C.

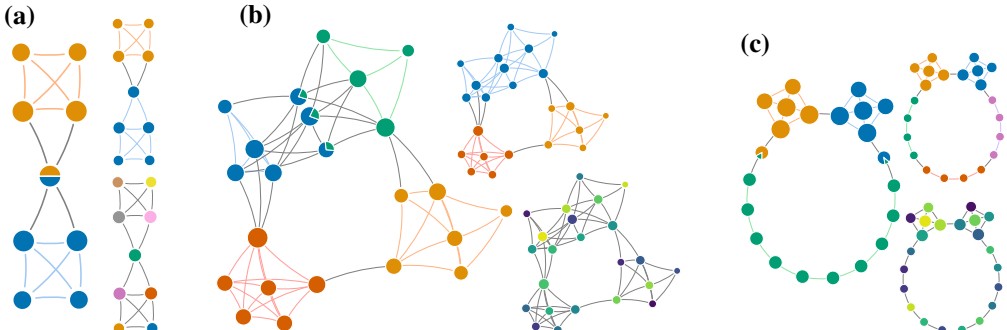

Figure 2: Synthetic networks with overlapping communities **(a, b)** and constrained structure **(c)**. In each panel, the larger networks on the left show the communities detected with our approach. The small networks at the top right show the communities detected with Infomap and the small networks at the bottom right the communities detected with DMoN.

Infomap has been shown to split constrained structures, such as rings of cliques or chains of nodes, at arbitrary points because such a split minimises the codelength (Edler et al., 2022). However, this often leads to unexpected partitions: consider the network in Figure 2c where Infomap splits the chain into three communities. Our approach can assign the nodes in the chain to the same community and identify the chain's starting nodes as overlapping. This is because a random walker has a higher probability of visiting these nodes from the denser groups than from the chain.

We leave a rigorous study of overlapping communities and fine-tuning for different constrained structures for future work.

## 4.3 REAL NETWORKS

We benchmark Neuromap on 8 datasets and compare its performance against other methods measured by Tsitsulin et al. (2023), including k-means on only node features (k-m (feat)), and in embeddings by DeepWalk (k-m (DW)) (Perozzi et al., 2014) and Deep Graph Infomax (k-m (DGI)) (Veličković et al., 2018), the stochastic block model (SBM) (Peixoto, 2014), Structural Deep Clustering Network (SDCN) (Bo et al., 2020), Attributed Graph Clustering (AGC) (Zhang et al., 2019). DAEGC (Wang et al., 2019), NOCD (Shchur & Günnemann, 2019), DiffPool (Ying et al., 2018), MinCut and Ortho (Bianchi et al., 2020), and DMoN (Tsitsulin et al., 2023). All 8 datasets were sourced from PyTorch Geometric (PyG) (Fey & Lenssen, 2019) and the Open Graph Benchmark (OGB) (Hu et al., 2020), consisting of small networks of around 3,000 nodes to large networks of greater than 100,000 nodes.

Table 1: Summary statistics of the benchmarked real-world datasets. Datasets were sourced via libraries PyTorch Geometric (PyG) Fey & Lenssen (2019) and Open Graph Benchmark (OGB) Hu et al. (2020) and thus adopted their naming. For brevity, we refer to the (Amazon) Computer and `ogbn-arxiv` datasets as "PC" and "arXiv", respectively.

| Dataset | Source | $|V|$ | $|E|$ | $|\mathbb{X}|$ | $|\mathbb{Y}|$ |
|---|---|---|---|---|---|
| (Planetoid) Cora | PyG and Sen et al. (2008) | 2708 | 5278 | 1433 | 7 |
| (Planetoid) CiteSeer | PyG and Sen et al. (2008) | 3327 | 4614 | 3703 | 6 |
| (Planetoid) Pubmed | PyG and Sen et al. (2008) | 19717 | 44325 | 500 | 3 |
| (Amazon) Computer | PyG and Shchur et al. (2018) | 13752 | 143604 | 767 | 10 |
| (Amazon) Photo | PyG and Shchur et al. (2018) | 7650 | 71831 | 745 | 8 |
| (Coauthor) CS | PyG and Shchur et al. (2018) | 18333 | 81894 | 6805 | 15 |
| (Coauthor) Physics | PyG and Shchur et al. (2018) | 34493 | 247962 | 8415 | 5 |
| `ogbn-arxiv` | OGB (Hu et al., 2020) | 169343 | 583121 | 128 | 40 |

Following Tsitsulin et al. (2023), for all datasets we use a graph neural network with 1 hidden layer, 512 hidden neurons, and 16 out channels, predicting a maximum of 16 clusters. We also use a similar graph neural network architecture using SeLU activation (Klambauer et al., 2017) and a trainable "skip-connection" instead of adding self-loops.

$$X_{\text{hidden}} = \text{SeLU}\left(A X_{\text{in}} W_{\text{encoder}} + X_{\text{in}} W_{\text{skip, encoder}}\right) \quad (9)$$

$$X_{\text{out}} = \text{softmax}\left(A X_{\text{hidden}} W_{\text{decoder}} + X_{\text{hidden}} W_{\text{skip, decoder}}\right) \quad (10)$$

We calculate the cluster assignment matrix as $S = X_{\text{out}} + \epsilon$ and set $\epsilon = 10^{-8}$ to ensure differentiability through the logarithms in the map equation loss function, as we did for our synthetic data benchmarks. We use a dropout layer before the softmax with dropout probability set to 0.5, set the maximum number of epochs to 10,000 for each run, and implement early stopping with a patience of 100 epochs.

For each dataset, we run Infomap for 100 trials to search for the partition that returns the lowest codelength. The map equation loss also guides tuning Neuromap's hyperparameters fairly in the unsupervised setting, again assuming that a lower codelength corresponds to a more accurate clustering of data. The learning rate is the only hyperparameter tuned for by searching the space $\{0.1, 0.05, 0.025, \ldots, 0.1 \times 0.5^{20-1}\}$ to find the learning rate which returns the lowest average codelength – that is, loss – over 10 runs, among which we select the result from the run producing the lowest codelength. We report additional experimental details and results on node features, learning rates, means, and standard deviations in Appendix D.

Neuromap performs competitively against other graph clustering benchmarks measured by Tsitsulin et al. (2023) in terms of NMI, including both deep learning and non-deep learning-based approaches (see Table 2). Among deep graph clustering approaches, Neuromap is among a select few able to converge on the Amazon PC and Amazon Photo datasets, where MinCut and Ortho were not able to. Tsitsulin et al. (2023)'s experiments involve a choice of 16 maximum clusters despite some datasets having a larger number of ground-truth clusters, e.g. the `ogbn-arxiv` dataset has 40 ground-truth clusters. As shown in our results on overlapping communities, DMoN has limitations in optimally determining the number of clusters and in identifying reasonable overlapping communities. When Neuromap is allowed a higher maximum number of clusters, specifically 100 and 1000, it demonstrates competitive performance on the `ogbn-arxiv` dataset, among others, underlining its proficiency in inferring a meaningful number of clusters without over-partitioning, a notable improvement over many existing methods. Additionally, in both synthetic and real networks, we observed that while Infomap attains lower codelengths, it does so at the cost of a significantly higher number of communities compared to Neuromap. This finding underscores that a low codelength alone is not indicative of meaningful communities, as it may misinterpret random patterns in sparse network areas as communities, a type of overfitting Neuromap successfully avoids.

We contribute these empirical insights while leaving further theoretical analysis for exciting future work. We also note that, while the benchmarks here do not have edge features, in principle edge features can be used for clustering with Neuromap.

Table 2: Normalised mutual information (NMI) scores for various methods on real-world networks (higher is better). Learnt numbers of clusters are reported for Infomap and Neuromap in parentheses. The best and second-best NMI scores, and closest and second-closest inferred cluster numbers to the ground truth, are highlighted in bold and underlined font, respectively, for Infomap and Neuromap. Standard deviations are reported in Table 6 in Appendix D.

| | Cora | CiteSeer | PubMed | PC | Photo | CS | Physics | arXiv |
|---|---|---|---|---|---|---|---|---|
| k-m (feat) | 18.5 | 24.5 | 19.4 | 21.1 | 28.8 | 35.7 | 30.6 | 20.3 |
| SBM | 36.2 | 15.3 | 16.4 | 48.4 | 59.3 | 58.0 | 45.4 | 31.9 |
| k-m (DW) | 24.3 | 27.6 | 22.9 | 38.2 | 49.4 | **72.7** | 43.5 | 28.4 |
| k-m (DGI) | **52.7** | **40.4** | 22.0 | 22.6 | 33.4 | 64.6 | 51.0 | 30.0 |
| SDCN | 27.9 | 31.4 | 19.5 | 24.9 | 41.7 | 59.3 | 50.4 | 15.3 |
| AGC | 34.1 | 25.5 | 18.2 | **51.3** | 59.0 | 43.3 | - | - |
| DAEGC | 8.3 | 4.3 | 4.4 | 42.5 | 47.6 | 59.3 | - | - |
| NOCD | 46.3 | 20.0 | 25.5 | 44.8 | 62.3 | 70.5 | 51.9 | 20.7 |
| DiffPool | 32.9 | 20.0 | 20.2 | 22.1 | 35.9 | 41.6 | - | - |
| MinCut | 35.8 | 25.9 | 25.4 | - | - | 64.6 | 48.3 | 36.0 |
| Ortho | 38.4 | 26.1 | 20.3 | - | - | 64.6 | 44.7 | 35.6 |
| DMoN | 48.8 | 33.7 | **29.8** | 49.3 | **63.3** | 69.1 | 56.7 | 37.6 |
| Infomap | 41.4 | 33.2 | 17.1 | 51.5 | 58.5 | 45.3 | 28.1 | **40.1** |
| | (289) | (628) | (939) | (458) | (221) | (824) | (1194) | (4808) |
| Neuromap | 49.3 | 22.0 | 26.6 | 34.3 | 56.0 | 69.7 | **58.5** | 28.7 |
| $(s = 16)$ | (**12**) | (**16**) | (**9**) | (**15**) | (**16**) | (**16**) | (**14**) | (5) |
| Neuromap | 47.9 | 23.9 | 23.3 | 36.1 | 53.6 | 64.9 | 47.6 | 32.4 |
| $(s = 100)$ | (45) | (57) | (37) | (54) | (39) | (46) | (45) | (7) |
| Neuromap | 46.2 | 25.5 | 21.7 | 42.2 | 57.3 | 65.9 | 43.8 | 35.6 |
| $(s = 1000)$ | (50) | (85) | (48) | (162) | (99) | (49) | (75) | (**15**) |

## 5 CONCLUSION

Network science and deep learning on graphs tackle community detection and graph clustering from different perspectives. Community detection in network science does not leverage recent advances in deep learning, and current deep graph clustering approaches have only recently started to incorporate methods from network science to improve clustering performance. We narrow this gap by adapting the map equation, a popular information-theoretic community-detection approach, as a differentiable loss function for optimisation with graph neural networks through gradient descent.

We applied our approach to various synthetic and real-world unsupervised graph clustering benchmarks, using different GNN architectures to minimise the map equation and detect communities. In real-world benchmarks, our approach achieves competitive performance, and synthetic benchmarks show that the quality of detected communities depends on the expressive power of the employed GNN. Through soft cluster assignments and optimisation with gradient descent, our approach naturally addresses over-partitioning in sparse networks, detecting overlapping communities, and avoiding splitting constrained structures.

Our adaptation of the map equation as a differentiable loss function for graph clustering with graph neural networks opens up several avenues for future research. Though our main focus was on unsupervised graph clustering for undirected networks, we also demonstrated how our approach can be used for directed networks where our results showed that further work is required to improve clustering performance and scalability. While we have considered first-order networks with two-level community structures, complex real-world networks often involve higher-order dependencies and can have multi-level communities (Rosvall & Bergstrom, 2011; Rosvall et al., 2014), prompting a generalisation of our approach. Our results highlight that the achieved cluster quality depends on the expressiveness of the GNN architecture used, but understanding the precise connection between the two requires further empirical and theoretical investigation. Although we expect our method to be easily incorporated into graph pooling (Tsitsulin et al., 2023) and scaling graph neural networks (Chiang et al., 2019), they require detailed further empirical and theoretical studies.

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

## A    BACKGROUND ON THE MAP EQUATION

We illustrate the principles behind the map equation using a coding example. Consider a communication game where the sender updates the receiver about the position of a random walker on a network. Both the sender and receiver know the network topology and use Huffman coding (Huffman, 1952) to assign unique codewords to nodes based on their stationary visit rates. When the random walker takes a step, the sender communicates the codeword of the random walker's current node to the receiver (Figure 3). With all nodes in the same module, the sender needs to use

$$L\left(\mathsf{M}_1\right) = H\left(P\right) = \sum_{u \in V} p_u \log_2 p_u \approx 3.07 \text{ bits} \tag{11}$$

on average to update the receiver about the random walker's position. Here, $p_u$ is the visit rate of node $u$, $P$ is the set of node visit rates, and $\mathsf{M}_1$ is the so-called one-level partition. In undirected graphs, we compute the nodes' visit rates directly as $p_u = s_u / \sum_{v \in V} s_v$, where $s_u = \sum_{v \in V} \delta\left(u, v\right)$ is the strength of node $u$. In directed graphs, we use smart teleportation (Lambiotte & Rosvall, 2012) and a power iteration to compute node visit rates numerically.

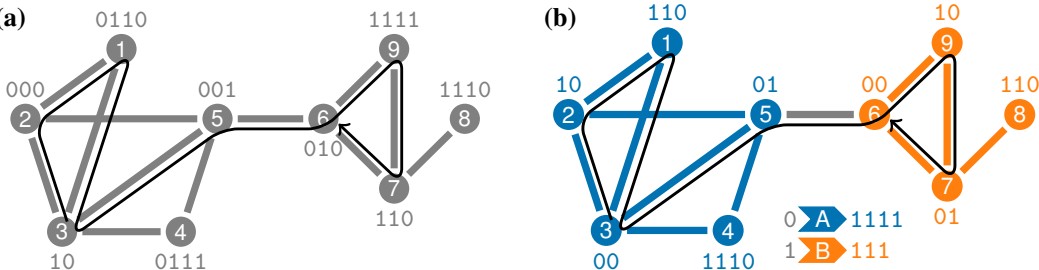

Figure 3: Illustration of the coding principles behind the map equation on an undirected network. Colours indicate modules and node codewords are shown next to nodes. The black trace shows a possible sequence of random-walker steps. Figure adapted from Blöcker (2022). **(a)** When all nodes belong to the same community, each node has a unique codeword. The random walk sequence is encoded as `10 000 0110 10 001 010 1111 110 010`, requiring 27 bits. **(b)** Partitioning the network into two communities enables reusing codewords for different nodes in different communities, reducing the overall codelength. However, for a unique encoding, we need to introduce codewords for describing when the random walker enters and exits modules as shown next to the coloured arrows. The random walk sequence is encoded as `000 10 110 00 01 1111` `00 10` `01 00`, requiring 25 bits, where colours indicate which codebook each codeword belongs to.

In networks with community structure, we can compress the description of the random walk by partitioning the nodes according to where the random walker tends to spend a relatively longer time (Figure 3b). We assign unique codewords to nodes per module, derived from their module-normalised visit rates; the same codeword can be used for nodes in different modules, overall making codewords shorter. The codelength is then a weighted average of the module-level entropies. However, we need to introduce designated module-exit codewords per module and an index-level codebook to describe when the random walker leaves and enters modules.

The two-level map equation calculates the codelength for a two-level partition $\mathsf{M}$,

$$L\left(\mathsf{M}\right) = q H\left(Q\right) + \sum_{\mathsf{m} \in \mathsf{M}} p_\mathsf{m} H\left(P_\mathsf{m}\right) \approx 2.62 \text{ bits.} \tag{12}$$

Here, $q = \sum_\mathsf{m} q_\mathsf{m}$ is the random walker's usage rate for the index-level codebook, $q_\mathsf{m}$ is the rate at which the random walker enters module $\mathsf{m}$, and $Q = \{q_\mathsf{m} \mid \mathsf{m} \in \mathsf{M}\}$ is the set of module entry rates; $p_\mathsf{m}$ is the rate at which the random walker uses module $\mathsf{m}$'s codebook, including module exit rate $\mathsf{m}_{\text{exit}}$, and $P_\mathsf{m} = \{\mathsf{m}_{\text{exit}}\} \cup \{p_u \mid u \in \mathsf{m}\}$ is the set of module-normalised node visit rates and exit rate for module $\mathsf{m}$.

# B   ADDITIONAL RESULTS ON LFR NETWORKS

We provide further results on the performance in undirected (Figure 4) and directed (Figure 5) LFR networks with planted community structure. We find that the results are qualitatively similar to what we described in the main text. $GNN_2$ performs best in small undirected networks. In larger undirected networks, Infomap performs slightly better. In directed networks, Infomap achieves higher NMI values than the tested neural network architectures. In denser networks and for low to medium mixing values $\mu$, all tested methods, except for $GNN_1$ detect communities that are in agreement with the ground truth.

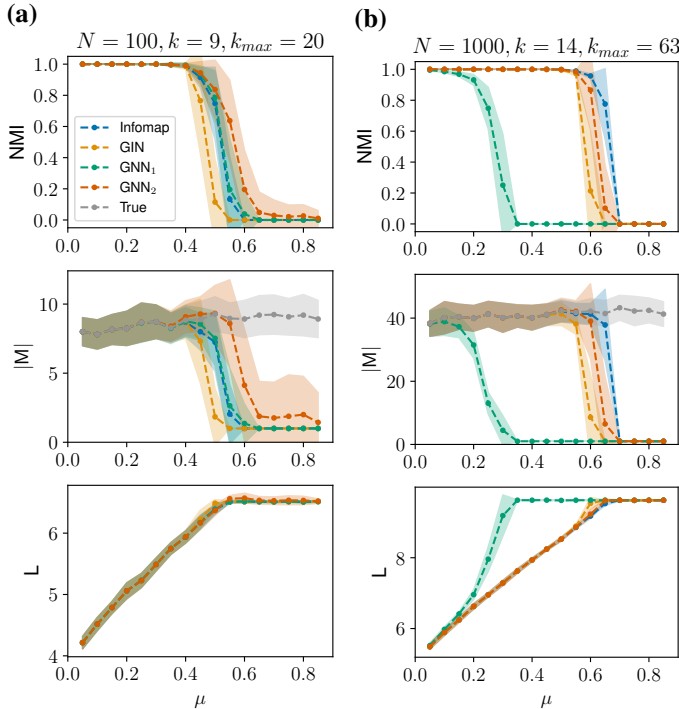

Figure 4: Results on denser synthetic undirected networks with $k = 2 \ln N$, rounded to the nearest integer, for Infomap, GIN, $GNN_1$, and $GNN_2$. Results are averages for partition quality in terms of NMI, number of detected communities $|M|$, and codelength $L$. Error bands show one standard deviation from the mean.

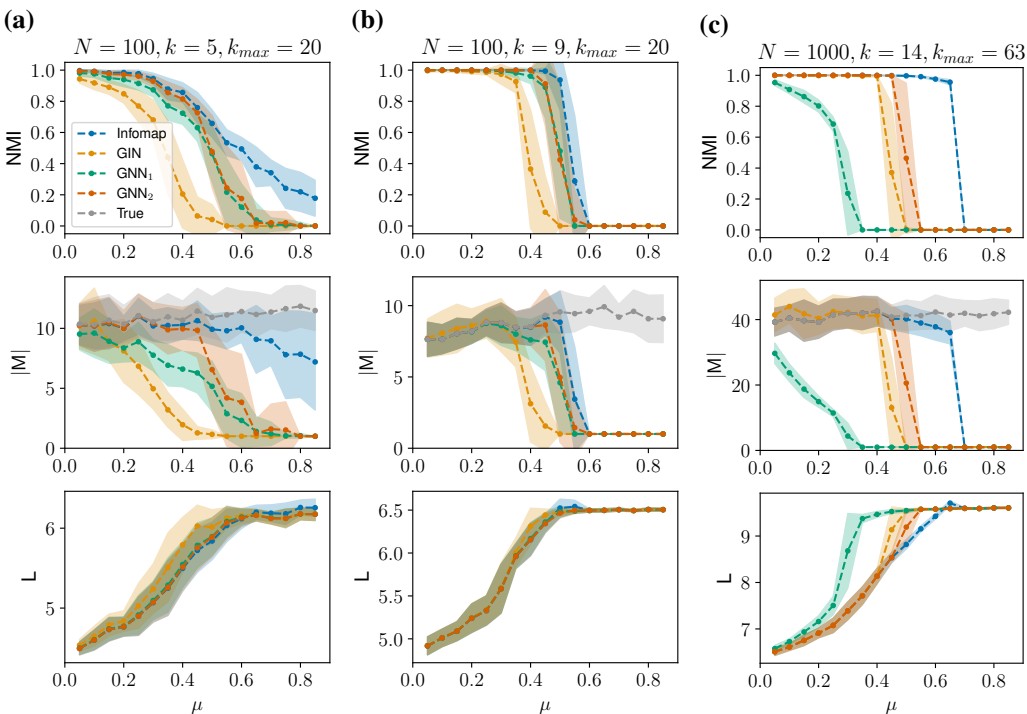

Figure 5: Results on directed synthetic networks for Infomap, GIN, GNN$_1$, and GNN$_2$. Results are averages for partition quality in terms of NMI, number of detected communities $|\mathsf{M}|$, and codelength $L$. Error bands show one standard deviation from the mean.

## C  ADDITIONAL NETWORKS WITH OVERLAPPING COMMUNITIES

We include further examples of networks with overlapping communities, specifically, synthetic LFR networks with planted community structures and a configurable number of nodes that belong to more than one community. The networks shown here have 50 nodes of which 5 belong to two communities and mixing of $\mu = 0.1$. We use GIN and a $\text{GNN}_1$, and run them for 10 trials to detect communities; both recover the ground truth communities due to low mixing and identify some of the overlapping nodes correctly.

**(a)**

**(b)**

**(c)**

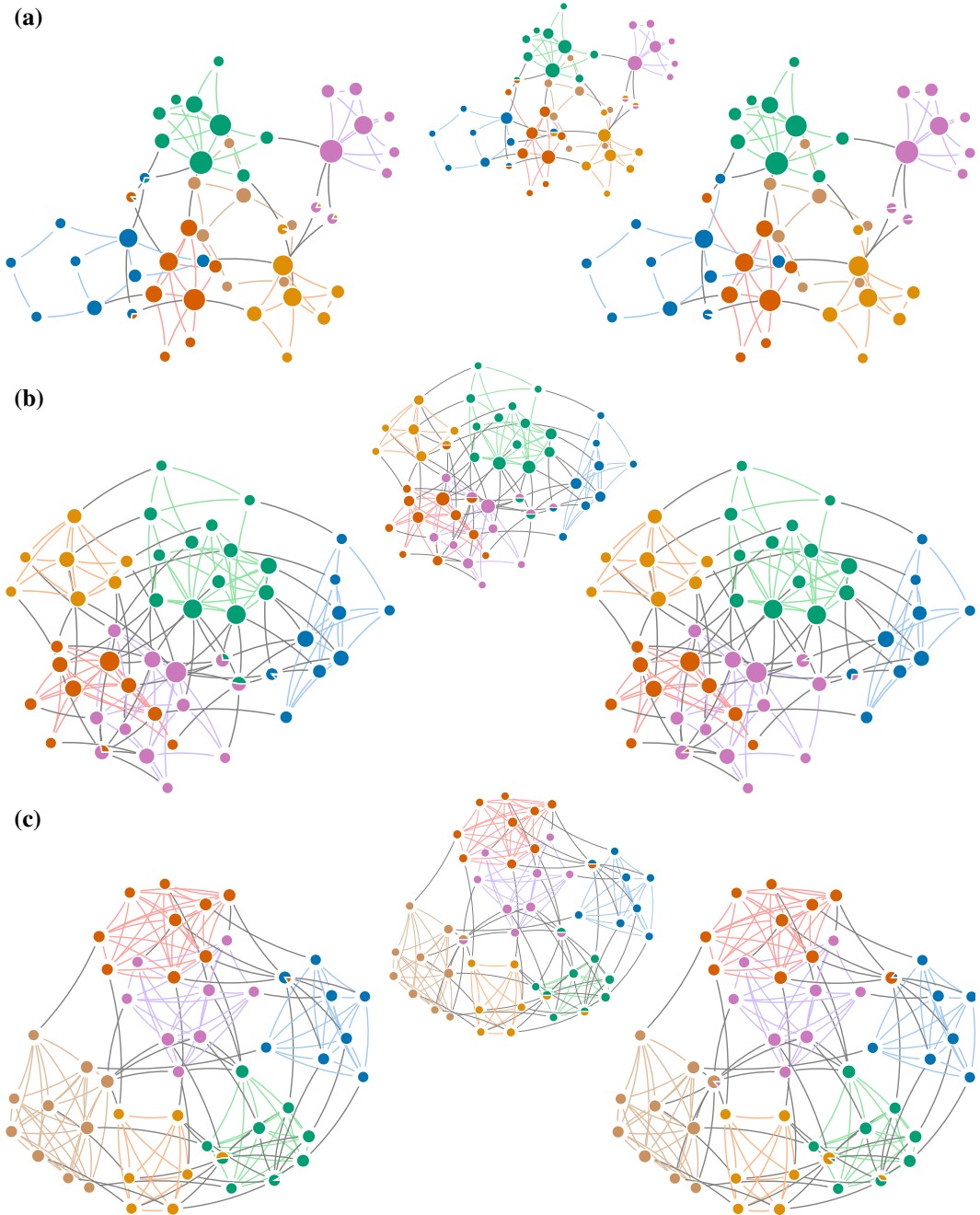

Figure 6: Networks with overlapping communities. Each panel shows three networks: the left-hand network is clustered using a GIN, the right-hand network is clustered using $\text{GNN}_1$, and the smaller middle network shows the ground truth communities.

# D    ADDITIONAL DETAILS AND RESULTS ON REAL NETWORKS

Here we provide additional experimental details and results on our real-world data benchmarks, including the learning rate chosen from hyperparameter tuning, as described in Section 4.3, in Table 3, modularity scores in Table 10, and means and standard deviations for normalised mutual information (NMI) in Tables 5 and 6, map equation codelengths in Tables 8 and 9, and modularity in Tables 10 to 12. Note that while the standard deviations for NMI and modularity are relatively large, the standard deviations of codelengths remain relatively small due to the complex loss landscape. However, as detailed in Section 4.3, even in the unsupervised setting, it is simple and theoretically principled to select the result that returns the lowest codelength from any number of trials.

We also test the performance of low dimensional ($\ll n$) positional encodings which maintain linear complexity for GNNs when training on large sparse graphs, simulating scenarios where real node features are either absent or inexpressive. We select 16 Laplacian eigenvector positional encodings, using the implementation in PyG guided by Dwivedi et al. (2020). We find that Neuromap learns accurate clusters from these positional encodings in some real-world networks, but performs poorly in most others; which we suspect is due to when Laplacian eigenvectors do not sufficiently capture graph topology, as evidenced by Infomap still performing well. Currently, designing expressive node features for graph neural networks is an active research area (Lim et al., 2022), and we expect more expressive node features to provide better performance in this setting.

Table 3: Learning rates for Neuromap on real-world networks.

| | | Cora | CiteSeer | PubMed | PC |
|---|---|---|---|---|---|
| Neuromap (feat+LE) | ($s = 16$) | 0.003125 | 0.003125 | 0.01250 | 0.000012 |
| Neuromap (feat) | ($s = 16$) | 0.001563 | 0.001563 | 0.00625 | 0.000002 |
| Neuromap (LE) | ($s = 16$) | 0.100000 | 0.100000 | 0.02500 | 0.003125 |
| Neuromap (feat+LE) | ($s = 100$) | 0.001563 | 0.001563 | 0.003125 | 0.000006 |
| Neuromap (feat) | ($s = 100$) | 0.001563 | 0.001563 | 0.006250 | 0.000012 |
| Neuromap (LE) | ($s = 100$) | 0.006250 | 0.012500 | 0.012500 | 0.000391 |
| Neuromap (feat+LE) | ($s = 1000$) | 0.001563 | 0.000781 | 0.003125 | 0.000006 |
| Neuromap (feat) | ($s = 1000$) | 0.001563 | 0.001563 | 0.003125 | 0.000006 |
| Neuromap (LE) | ($s = 1000$) | 0.012500 | 0.000781 | 0.006250 | 0.100000 |
| | | Photo | CS | Physics | arXiv |
| Neuromap (feat+LE) | ($s = 16$) | 0.000006 | 0.000195 | 0.001563 | 0.000195 |
| Neuromap (feat) | ($s = 16$) | 0.000012 | 0.001563 | 0.001563 | 0.000391 |
| Neuromap (LE) | ($s = 16$) | 0.003125 | 0.025000 | 0.025000 | 0.100000 |
| Neuromap (feat+LE) | ($s = 100$) | 0.000006 | 0.000195 | 0.000781 | 0.000391 |
| Neuromap (feat) | ($s = 100$) | 0.000012 | 0.000391 | 0.000781 | 0.000781 |
| Neuromap (LE) | ($s = 100$) | 0.001563 | 0.006250 | 0.012500 | 0.012500 |
| Neuromap (feat+LE) | ($s = 1000$) | 0.000012 | 0.000391 | 0.000781 | 0.000049 |
| Neuromap (feat) | ($s = 1000$) | 0.000006 | 0.000391 | 0.000781 | 0.000098 |
| Neuromap (LE) | ($s = 1000$) | 0.012500 | 0.006250 | 0.006250 | 0.100000 |

Table 4: Normalised mutual information (NMI) scores for various methods on real-world networks (higher is better). Inferred numbers of clusters are reported in parentheses. For Neuromap, "feat" refers to using real node features, and "LE" refers to using Laplacian eigenvector positional encodings as node features. Standard deviations are reported in Table 6.

|  | Cora | CiteSeer | PubMed | PC | Photo | CS | Physics | arXiv |
|---|---|---|---|---|---|---|---|---|
| Neuromap (feat+LE) | 45.3 | 22.8 | 28.0 | 27.0 | 51.8 | 74.0 | 55.1 | 28.7 |
| ($s = 16$) | (11) | (13) | (7) | (16) | (11) | (15) | (14) | (7) |
| Neuromap (LE) | 2.2 | 0.2 | 24.5 | 13.4 | 38.0 | 56.0 | 53.1 | 0.1 |
| ($s = 16$) | (7) | (1) | (6) | (2) | (11) | (10) | (10) | (4) |
| Neuromap (feat+LE) | 47.6 | 24.4 | 23.2 | 43.6 | 58.2 | 67.2 | 48.1 | 30.3 |
| ($s = 100$) | (36) | (67) | (39) | (51) | (44) | (41) | (44) | (7) |
| Neuromap (LE) | 22.6 | 3.7 | 21.9 | 40.1 | 45.8 | 58.1 | 57.5 | 0.1 |
| ($s = 100$) | (11) | (14) | (12) | (9) | (9) | (16) | (17) | (4) |
| Neuromap (feat+LE) | 48.6 | 25.2 | 23.0 | 42.3 | 60.0 | 65.2 | 44.0 | 32.7 |
| ($s = 1000$) | (39) | (97) | (38) | (142) | (91) | (54) | (77) | (17) |
| Neuromap (LE) | 11.9 | 0.0 | 23.2 | 0.1 | 53.2 | 60.2 | 53.2 | 0.3 |
| ($s = 1000$) | (12) | (1) | (18) | (2) | (14) | (22) | (13) | (8) |

Table 5: Mean normalised mutual information (NMI) scores for Neuromap on real-world networks (higher is generally better).

|  |  | Cora | CiteSeer | PubMed | PC | Photo | CS | Physics | arXiv |
|---|---|---|---|---|---|---|---|---|---|
| $s = 16$ | feat+LE | 43.5 | 22.0 | 23.3 | 10.0 | 50.6 | 71.7 | 56.1 | 26.1 |
| $s = 16$ | feat | 44.8 | 23.6 | 24.4 | 28.1 | 50.0 | 68.9 | 55.4 | 24.8 |
| $s = 16$ | LE | 0.5 | 0.3 | 16.4 | 7.6 | 34.3 | 38.5 | 37.4 | 0.1 |
| $s = 100$ | feat+LE | 47.7 | 24.7 | 23.4 | 37.6 | 55.6 | 67.2 | 48.1 | 29.2 |
| $s = 100$ | feat | 47.3 | 24.7 | 23.5 | 34.8 | 55.6 | 66.8 | 48.2 | 29.8 |
| $s = 100$ | LE | 13.2 | 0.5 | 20.6 | 33.5 | 44.7 | 53.7 | 52.4 | 0.1 |
| $s = 1000$ | feat+LE | 47.6 | 26.0 | 23.2 | 38.6 | 57.8 | 66.6 | 46.3 | 30.4 |
| $s = 1000$ | feat | 47.4 | 25.1 | 23.0 | 38.6 | 56.6 | 66.4 | 46.1 | 31.1 |
| $s = 1000$ | LE | 8.0 | 0.0 | 21.3 | 0.2 | 54.9 | 58.4 | 54.2 | 0.3 |

Table 6: Standard deviation of normalised mutual information (NMI) scores for Neuromap on real-world networks.

|  |  | Cora | CiteSeer | PubMed | PC | Photo | CS | Physics | arXiv |
|---|---|---|---|---|---|---|---|---|---|
| $s = 16$ | feat+LE | 3.7 | 2.9 | 5.3 | 11.5 | 5.0 | 1.5 | 2.1 | 3.9 |
| $s = 16$ | feat | 2.9 | 2.5 | 3.0 | 8.6 | 4.5 | 2.0 | 2.2 | 2.8 |
| $s = 16$ | LE | 0.7 | 0.4 | 11.6 | 6.9 | 4.0 | 20.3 | 26.0 | 0.0 |
| $s = 100$ | feat+LE | 1.7 | 0.9 | 1.1 | 3.5 | 3.7 | 1.9 | 0.8 | 2.7 |
| $s = 100$ | feat | 1.9 | 1.0 | 0.8 | 10.7 | 3.7 | 1.2 | 0.9 | 2.2 |
| $s = 100$ | LE | 6.2 | 1.1 | 7.3 | 5.3 | 0.9 | 8.2 | 2.2 | 0.0 |
| $s = 1000$ | feat+LE | 0.8 | 0.9 | 0.6 | 3.3 | 2.4 | 0.8 | 1.3 | 1.7 |
| $s = 1000$ | feat | 2.0 | 1.4 | 1.0 | 3.5 | 2.2 | 1.4 | 1.2 | 2.8 |
| $s = 1000$ | LE | 4.5 | 0.0 | 2.7 | 0.2 | 1.3 | 1.2 | 1.6 | 0.4 |

Table 7: Codelengths for map equation-based clustering methods on real-world networks (lower is generally better). For Neuromap, "feat" refers to using real node features, and "LE" refers to using Laplacian eigenvector positional encodings as node features. Standard deviations are reported in Table 9 in Appendix D.

| | Cora | CiteSeer | PubMed | PC | Photo | CS | Physics | arXiv |
|---|---|---|---|---|---|---|---|---|
| Infomap | 6.4 | 5.0 | 8.7 | 10.6 | 9.4 | 9.8 | 11.1 | 11.9 |
| Neuromap (feat+LE) ($s = 16$) | 11.2 | 11.4 | 13.8 | 13.7 | 12.7 | 14.0 | 15.0 | 16.6 |
| Neuromap (feat) ($s = 16$) | 11.3 | 11.4 | 13.7 | 13.7 | 12.7 | 14.0 | 15.0 | 16.7 |
| Neuromap (LE) ($s = 16$) | 12.0 | 12.2 | 13.9 | 13.8 | 12.9 | 14.2 | 15.2 | 17.0 |
| Neuromap (feat+LE) ($s = 100$) | 11.0 | 10.9 | 13.4 | 13.6 | 12.6 | 13.8 | 14.8 | 16.7 |
| Neuromap (feat) ($s = 100$) | 11.0 | 11.0 | 13.5 | 13.6 | 12.6 | 13.8 | 14.8 | 16.7 |
| Neuromap (LE) ($s = 100$) | 11.8 | 11.9 | 13.8 | 13.6 | 12.8 | 14.1 | 15.1 | 17.0 |
| Neuromap (feat+LE) ($s = 1000$) | 10.9 | 10.8 | 13.4 | 13.6 | 12.6 | 13.8 | 14.8 | 16.6 |
| Neuromap (feat) ($s = 1000$) | 11.0 | 10.9 | 13.4 | 13.5 | 12.6 | 13.8 | 14.8 | 16.6 |
| Neuromap (LE) ($s = 1000$) | 11.8 | 12.2 | 13.7 | 13.9 | 12.6 | 14.1 | 15.1 | 17.0 |

Table 8: Mean codelengths for Neuromap on real-world networks (lower is generally better).

| | | Cora | CiteSeer | PubMed | PC | Photo | CS | Physics | arXiv |
|---|---|---|---|---|---|---|---|---|---|
| $s = 16$ | feat+LE | 11.4 | 11.5 | 13.8 | 13.8 | 12.9 | 14.0 | 15.0 | 16.7 |
| $s = 16$ | feat | 11.4 | 11.5 | 13.8 | 13.7 | 12.8 | 14.1 | 15.0 | 16.7 |
| $s = 16$ | LE | 12.0 | 12.2 | 14.0 | 13.8 | 12.9 | 14.4 | 15.3 | 17.0 |
| $s = 100$ | feat+LE | 11.1 | 11.1 | 13.5 | 13.6 | 12.7 | 13.8 | 14.8 | 16.7 |
| $s = 100$ | feat | 11.1 | 11.1 | 13.6 | 13.7 | 12.7 | 13.8 | 14.8 | 16.7 |
| $s = 100$ | LE | 11.8 | 12.2 | 13.9 | 13.6 | 12.8 | 14.2 | 15.1 | 17.0 |
| $s = 1000$ | feat+LE | 11.0 | 11.0 | 13.5 | 13.6 | 12.6 | 13.8 | 14.8 | 16.6 |
| $s = 1000$ | feat | 11.0 | 11.0 | 13.5 | 13.6 | 12.6 | 13.8 | 14.8 | 16.6 |
| $s = 1000$ | LE | 11.9 | 12.2 | 13.8 | 13.9 | 12.6 | 14.1 | 15.1 | 17.0 |

Table 9: Standard deviation of codelengths for Neuromap on real-world networks.

| | | Cora | CiteSeer | PubMed | PC | Photo | CS | Physics | arXiv |
|---|---|---|---|---|---|---|---|---|---|
| $s = 16$ | feat+LE | 0.1 | 0.1 | 0.1 | 0.0 | 0.1 | 0.0 | 0.0 | 0.1 |
| $s = 16$ | feat | 0.1 | 0.1 | 0.1 | 0.1 | 0.1 | 0.0 | 0.0 | 0.0 |
| $s = 16$ | LE | 0.0 | 0.0 | 0.2 | 0.0 | 0.0 | 0.2 | 0.2 | 0.0 |
| $s = 100$ | feat+LE | 0.1 | 0.1 | 0.1 | 0.0 | 0.0 | 0.0 | 0.0 | 0.0 |
| $s = 100$ | feat | 0.1 | 0.1 | 0.1 | 0.1 | 0.1 | 0.0 | 0.0 | 0.0 |
| $s = 100$ | LE | 0.1 | 0.1 | 0.2 | 0.0 | 0.0 | 0.2 | 0.0 | 0.0 |
| $s = 1000$ | feat+LE | 0.1 | 0.2 | 0.1 | 0.1 | 0.0 | 0.0 | 0.0 | 0.0 |
| $s = 1000$ | feat | 0.1 | 0.1 | 0.0 | 0.0 | 0.0 | 0.0 | 0.0 | 0.0 |
| $s = 1000$ | LE | 0.1 | 0.0 | 0.2 | 0.0 | 0.0 | 0.0 | 0.0 | 0.0 |

Table 10: Modularity scores for various methods on real-world networks (higher is generally better).

| | Cora | CiteSeer | PubMed | PC | Photo | CS | Physics | arXiv |
|---|---|---|---|---|---|---|---|---|
| k-m (feat) | 19.8 | 30.3 | 33.4 | 5.4 | 10.5 | 23.1 | 19.4 | 16.4 |
| SBM | 77.3 | 78.1 | 53.5 | 60.8 | 72.7 | 72.7 | 66.9 | 67.6 |
| k-m (DW) | 30.7 | 24.3 | 75.3 | 11.8 | 22.9 | 59.4 | 47.0 | 58.2 |
| k-m (DGI) | 64.0 | 73.7 | 9.6 | 22.8 | 35.1 | 57.8 | 51.2 | 29.7 |
| SDCN | 50.8 | 62.3 | 50.3 | 45.6 | 53.3 | 55.7 | 52.8 | 36.8 |
| AGC | 43.2 | 50.0 | 46.8 | 42.8 | 55.9 | 40.1 | - | - |
| DAEGC | 33.5 | 36.4 | 37.5 | 43.3 | 58.0 | 49.1 | - | - |
| NOCD | 78.3 | 84.4 | 69.6 | 59.0 | 70.1 | 72.2 | 65.5 | 41.9 |
| DiffPool | 66.3 | 63.4 | 56.8 | 30.4 | 46.8 | 41.6 | - | - |
| MinCut | 70.3 | 78.9 | 63.1 | - | - | 64.6 | 64.3 | 52.6 |
| Ortho | 65.6 | 74.5 | 32.9 | - | - | 64.6 | 59.5 | 52.2 |
| DMoN | 76.5 | 79.3 | 65.4 | 59.0 | 70.1 | 72.4 | 65.8 | 57.4 |
| Infomap | 73.1 | 82.8 | 65.8 | 59.4 | 70.6 | 59.6 | 52.6 | 59.3 |
| Neuromap (feat+LE) ($s = 16$) | 66.0 | 76.5 | 67.4 | 41.4 | 59.8 | 73.4 | 64.7 | 45.8 |
| Neuromap (feat) ($s = 16$) | 74.5 | 79.5 | 70.8 | 53.6 | 66.1 | 71.5 | 65.8 | 45.3 |
| Neuromap (LE) ($s = 16$) | 4.3 | 0.1 | 60.1 | 33.0 | 48.4 | 65.7 | 56.8 | 0.1 |
| Neuromap (feat+LE) ($s = 100$) | 77.1 | 83.5 | 71.6 | 53.6 | 66.1 | 72.3 | 65.5 | 44.8 |
| Neuromap (feat) ($s = 100$) | 76.6 | 80.3 | 69.4 | 50.0 | 64.1 | 71.5 | 64.7 | 46.5 |
| Neuromap (LE) ($s = 100$) | 42.0 | 34.0 | 64.4 | 49.4 | 52.7 | 66.4 | 58.1 | 0.1 |
| Neuromap (feat+LE) ($s = 1000$) | 75.4 | 85.0 | 71.2 | 50.1 | 63.9 | 70.1 | 63.2 | 56.7 |
| Neuromap (feat) ($s = 1000$) | 76.7 | 83.2 | 71.7 | 49.0 | 64.8 | 71.6 | 63.4 | 56.8 |
| Neuromap (LE) ($s = 1000$) | 20.0 | 0.0 | 66.4 | 0.0 | 59.7 | 67.4 | 59.2 | 0.1 |

Table 11: Mean modularity scores for Neuromap on real-world networks.

| | | Cora | CiteSeer | PubMed | PC | Photo | CS | Physics | arXiv |
|---|---|---|---|---|---|---|---|---|---|
| $s = 16$ | feat+LE | 69.9 | 77.3 | 64.2 | 13.7 | 59.8 | 73.0 | 64.6 | 44.8 |
| $s = 16$ | feat | 72.6 | 78.9 | 66.5 | 41.8 | 59.7 | 71.0 | 64.9 | 44.2 |
| $s = 16$ | LE | 0.6 | 1.7 | 38.9 | 16.4 | 47.1 | 47.4 | 37.7 | 0.1 |
| $s = 100$ | feat+LE | 76.1 | 82.5 | 70.8 | 46.0 | 64.5 | 71.5 | 64.6 | 44.0 |
| $s = 100$ | feat | 75.3 | 82.0 | 68.6 | 40.6 | 61.6 | 71.0 | 64.8 | 45.3 |
| $s = 100$ | LE | 23.6 | 3.6 | 54.6 | 46.1 | 52.3 | 62.6 | 56.8 | 0.0 |
| $s = 1000$ | feat+LE | 76.4 | 83.3 | 70.5 | 42.4 | 63.9 | 70.5 | 63.6 | 49.7 |
| $s = 1000$ | feat | 75.4 | 81.7 | 70.3 | 46.9 | 63.2 | 70.9 | 63.6 | 49.1 |
| $s = 1000$ | LE | 12.6 | 0.0 | 60.5 | 0.0 | 58.8 | 66.3 | 58.2 | 0.1 |

Table 12: Standard deviations of modularity scores for Neuromap on real-world networks.

|  |  | Cora | CiteSeer | PubMed | PC | Photo | CS | Physics | arXiv |
|---|---|---|---|---|---|---|---|---|---|
| $s = 16$ | feat+LE | 2.7 | 1.5 | 5.0 | 17.2 | 3.9 | 0.3 | 0.7 | 2.1 |
| $s = 16$ | feat | 1.5 | 1.7 | 2.8 | 14.1 | 4.2 | 0.7 | 0.8 | 1.6 |
| $s = 16$ | LE | 1.4 | 3.0 | 27.3 | 16.1 | 2.6 | 25.2 | 26.2 | 0.0 |
| $s = 100$ | feat+LE | 1.0 | 1.0 | 1.2 | 8.2 | 3.2 | 1.2 | 0.9 | 2.6 |
| $s = 100$ | feat | 1.4 | 1.1 | 1.7 | 14.8 | 4.8 | 0.7 | 0.3 | 0.8 |
| $s = 100$ | LE | 13.5 | 10.7 | 19.5 | 3.0 | 0.5 | 6.7 | 0.9 | 0.0 |
| $s = 1000$ | feat+LE | 0.7 | 1.6 | 1.4 | 9.9 | 3.0 | 0.5 | 1.0 | 5.1 |
| $s = 1000$ | feat | 1.5 | 1.4 | 0.9 | 5.3 | 3.1 | 0.7 | 0.7 | 5.9 |
| $s = 1000$ | LE | 8.8 | 0.0 | 9.0 | 0.0 | 1.5 | 1.0 | 1.1 | 0.0 |

