# OpenReview forum: "The Map Equation goes Neural"
_ICLR.cc/2024/Conference — Submitted to ICLR 2024_

### Official Review · Reviewer_r6gM · 2023-10-16

**Soundness:** 2 fair
**Presentation:** 3 good
**Contribution:** 3 good
**Rating:** 6
**Confidence:** 5

**Summary:**

The authors formulate the well-known MAP equation for community detection as an unsupervised objective for graph clustering with GNNs. The implement this "soft" neural MAP equation in various GNN architectures, showing reasonable performance on both synthetic and real-world graph clustering tasks.

**Strengths:**

S1: The port of the MAP equation to a NN graph clustering objective is good to have in the modern-day toolkit of neural clustering techniques.

S2: The paper is well-written and easy to follow.

S3: The experiments are sufficient and easy to understand.

**Weaknesses:**

W1: The contribution itself is marginal. The authors seem to simply replace the objective of Tsitsulin et al. 2023 with the MAP equation.

W2: The authors claim that the MAP equation avoids over-partitioning, but do not provide any theoretical justification.

W3: The authors claim the ability to detect overlapping communities as a contribution of their work, but this is also true of any "soft clustering" neural method including Tsitsulin et al. 2023.

**Questions:**

My questions are as follows:

(1) re W1, Can the authors claim any technical novelty beyond deriving the MAP equation as a neural objective and using the approach of Tsitsulin et al. 2023?

(2) re W2, on page 4, the authors claim "the map equation naturally incorporates Occam's razor: minimising the map equation requires a trade-off between choosing small modules for low module-level codelength and choosing a small number of modules for low index-level codelength".

This is a strong claim but no theoretical justification was given. It is not clear nor obvious how the Occam's razor concept can be rigorously formulated in (or satisfied by) a neural clustering objective. As was done in Tsitsulin et al. 2023, the authors should formally argue how their objective avoids the collapse condition (all nodes in singleton clusters or in the unity cluster).

(3) The authors claim that a contribution of their approach is the ability to return overlapping cluster assignments. However, this is true of any neural clustering method with soft clustering assignments, including that of Tsitsulin et al. 2023. Can the authors compare the results in Fig 2 with those obtained by DMoN? If those obtained by NeuroMAP appear better, intuitive explanation of the improvement should also be stated.

---

> ### Author Response · Authors · 2023-11-21
>
> > The authors formulate the well-known MAP equation for community detection as an unsupervised objective for graph clustering with GNNs. The implement this "soft" neural MAP equation in various GNN architectures, showing reasonable performance on both synthetic and real-world graph clustering tasks.
> >
> > Strenghts
> >
> > - The port of the MAP equation to a NN graph clustering objective is good to have in the modern-day toolkit of neural clustering techniques.
> > - The paper is well-written and easy to follow.
> > - The experiments are sufficient and easy to understand.
>
> We thank the reviewer for the feedback and for highlighting the relevance of our contribution to the modern-day toolkit of neural clustering.
>
> > **(1)** _W1: The contribution itself is marginal. The authors seem to simply replace the objective of Tsitsulin et al. 2023 with the MAP equation. re W1, Can the authors claim any technical novelty beyond deriving the MAP equation as a neural objective and using the approach of Tsitsulin et al. 2023?_
>
> We did not use the approach of Tsitsulin et al. 2023. The contribution and approach of Tsitsulin et al. 2023 was a neural clustering objective. Our work was inspired by DiffPool and, similar to Tsitsulin et al. (2023), we also propose an objective function for community detection with GNNs. As the results in Fig. 2 illustrate, the objectives of DMoN and Neurmap cannot simply be exchanged: using DMoN to cluster small synthetic networks without informative node features returns singleton clusters.
>
> Our work contributes to narrowing the gap between the network science and deep learning communities. As we explain, optimising the map equation with GNNs naturally addresses several issues of Infomap:
>
> _"[...] our approach naturally addresses over-partitioning in sparse networks, detecting overlapping communities, and avoiding splitting constrained structures."_
>
> > **(2)** _W2: The authors claim that the MAP equation avoids over-partitioning, but do not provide any theoretical justification. re W2, on page 4, the authors claim "the map equation naturally incorporates Occam's razor: minimising the map equation requires a trade-off between choosing small modules for low module-level codelength and choosing a small number of modules for low index-level codelength"._
> >
> > _This is a strong claim but no theoretical justification was given. It is not clear nor obvious how the Occam's razor concept can be rigorously formulated in (or satisfied by) a neural clustering objective. As was done in Tsitsulin et al. 2023, the authors should formally argue how their objective avoids the collapse condition (all nodes in singleton clusters or in the unity cluster)._
>
> Avoiding collapsing to a trivial solution where all nodes are assigned to the same cluster or each node is assigned to its own singleton cluster is achieved by following the minimum description length principle, an information-theoretic model-selection approach that has long been used to balance between model complexity and fit [1,2]. The map equation avoids over- and underfitting by following the minimum description length principles as extensively theoretically explored and discussed in previous work related to the map equation [3,4].
>
> [1] Rissanen, Jorma. "Modeling by shortest data description." Automatica 14.5 (1978): 465-471. \
> [2] Grünwald, Peter D., In Jae Myung, and Mark A. Pitt, eds. Advances in minimum description length: Theory and applications. MIT press, 2005. \
> [3] Rosvall, Martin, Daniel Axelsson, and Carl T. Bergstrom. "The map equation." The European Physical Journal Special Topics 178.1 (2009): 13-23. \
> [4] Rosvall, Martin, and Carl T. Bergstrom. "Maps of random walks on complex networks reveal community structure." Proceedings of the national academy of sciences 105.4 (2008): 1118-1123.

---

> ### Author Response · Authors · 2023-11-21
>
> > **(3)** _W3: The authors claim the ability to detect overlapping communities as a contribution of their work, but this is also true of any "soft clustering" neural method including Tsitsulin et al. 2023. The authors claim that a contribution of their approach is the ability to return overlapping cluster assignments. However, this is true of any neural clustering method with soft clustering assignments, including that of Tsitsulin et al. 2023. Can the authors compare the results in Fig 2 with those obtained by DMoN? If those obtained by NeuroMAP appear better, intuitive explanation of the improvement should also be stated._
>
> Our work enables detecting overlapping clusters from first-order data with Neuromap whereas the standard map equation requires a higher-order model as we explain in section 4.2.:
>
> _"Through soft cluster assignment, we naturally discover overlapping communities (Figure 2a-b). In contrast, the standard map equation with hard cluster assignments requires higher-order data (Rosvall et al., 2014) or flow modelling (Holmgren et al., 2023) and a memory network representation to detect overlapping communities"_
>
> We have clustered the networks in Fig. 2 with DMoN and added the results to Fig. 2. Because the nodes in these networks have no features, we use the identity matrix as node features. We find that DMoN returns each node in its own singleton community, suggesting that DMoN focuses on node features rather than topological patterns.

---

> > ### Comment · Reviewer_r6gM · 2023-11-23
> > **Modified score**
> >
> > Thanks to the authors for their time in replying. I agree that the existing literature on the MAP equation provides a bit more justification for its use than I originally realized. I also agree that the contribution goes further beyond Tsitsulin et al. than I originally realized. For these reasons I have raised my "contribution" score by 1 and the overall rating by 2.

---

> > > ### Author Response · Authors · 2023-11-23
> > >
> > > > Thanks to the authors for their time in replying. I agree that the existing literature on the MAP equation provides a bit more justification for its use than I originally realized. I also agree that the contribution goes further beyond Tsitsulin et al. than I originally realized. For these reasons I have raised my "contribution" score by 1 and the overall rating by 2.
> > >
> > > We sincerely thank the reviewer for their time and effort going through our revisions and replies, and for acknowledging the value of our work, considering it worthy of publication in ICLR.

---

### Official Review · Reviewer_rHRz · 2023-10-25

**Soundness:** 1 poor
**Presentation:** 1 poor
**Contribution:** 2 fair
**Rating:** 1
**Confidence:** 4

**Summary:**

This paper proposes a new community detection algorithm based on the map equation that is the objective function of the well-known Infomap algorithm (Rosvall and Bergstrom, 2008). It treats the map equation as the (differentiable) loss function of graph neural networks for hard and soft clustering. Experimental results demonstrate the effectiveness of this method.

**Strengths:**

The idea of combining an information-theoretic cost function for clustering with neural networks is new.

**Weaknesses:**

(1) The presentation of this paper is quite poor. The notations in the Map Equation Loss section, which is the most significant part of this paper, are totally confusing.

(2) The description of Neuromap is too compressed. The details of GNNs with the map equation loss are missing.

(3) The experimental results are not convincing. The results of Neuromap in Figure 1, Tables 2 and 3 are hard to say competitive. It seems that the original Infomap algorithm performs better on many benchmarks.

**Questions:**

(1) In the Map Equation Loss section, what does the boldface $\textbf{A}_{i,j}$ mean? Is $\textbf{p}$ a vector or matrix? What is the definition of flow matrix? What does $\propto$ mean?

(2) Can you provide more details on the neural networks?

(3) How do you identify the overlapping communities in your algorithm?

(4) What is the efficiency of Neuromap in the experiments?

---

> ### Author Response · Authors · 2023-11-21
>
> > This paper proposes a new community detection algorithm based on the map equation that is the objective function of the well-known Infomap algorithm (Rosvall and Bergstrom, 2008). It treats the map equation as the (differentiable) loss function of graph neural networks for hard and soft clustering. Experimental results demonstrate the effectiveness of this method.
> >
> > The idea of combining an information-theoretic cost function for clustering with neural networks is new.
>
> We thank the reviewer for providing feedback and highlighting the novely of our approach.
>
> > **(1)** _The presentation of this paper is quite poor. The notations in the Map Equation Loss section, which is the most significant part of this paper, are totally confusing._
>
> Thank you for pointing out that the notation may be a hindrance for understanding our work. We have revised the mathematical notation, adopting Einstein notation for greater clarity of the rank of tensors, including the differences between matrices and vectors.
>
> > **(2)** _The description of Neuromap is too compressed. The details of GNNs with the map equation loss are missing._
>
> We have revised our notation for clarity. For details about the GNNs, we refer the reviewer to section 4.1. and 4.3. where the GNN architectures are specified.
>
> > **(3)** _The experimental results are not convincing. The results of Neuromap in Figure 1, Tables 2 and 3 are hard to say competitive. It seems that the original Infomap algorithm performs better on many benchmarks._
>
> Infomap achieves lower codelengths and higher NMI values especially for high mixing values. However, examining the number of detected communities shows that Infomap is overfitting and detects considerably more communities than are present in the ground truth. For low mixing values, Figure 1 shows that Neuromap achieves similar codelengths and NMI values as Infomap, even slightly outperforming Infomap. Importantly, achieving a low codelength alone does not guarantee that the identified modules are meaningful because the map equation tends to detect spurious modules in sparse network regions [1].
>
> As the revised version of table 2 and 7 shows, we observe a similar behaviour in real networks where Infomap achieves lower codelengths at the expense of a considerably higher number of communities than Neuromap. Again, a low codelength alone does not guarantee meaningful communities because random patterns in sparse network regions appear as though they are communities -- Neuromap avoids this overfitting.
>
> [1] Smiljanić et al., Mapping flows on weighted and directed networks with incomplete observations
>
> ### Questions:
>
> > **(1)** _In the Map Equation Loss section, what does the boldface $\mathbf{A}\_{i, j}$ mean? Is a $\mathbf{p}$ a vector or matrix? What is the definition of flow matrix? What does $\propto$ mean?_
>
> We have revised our mathematical notation for better clarity.
>
> - $\mathbf{p}$ is a vector whose elements are the node's visit rates.
> - The flow matrix specifies the fraction of flow on each link in the network, that is, the probability that a random walker uses each respective link. We have added an explanation to the manuscript.
> - $\propto$ is the mathematical symbol to denote proportionality. However, to avoid confusion, we have removed the symbol from the manuscript.
>
> > **(2)** _Can you provide more details on the neural networks?_
>
> Please refer to section 4.1. and 4.3. where we have detailed the GNNs' architectures.
>
> > **(3)** _How do you identify the overlapping communities in your algorithm?_
>
> We detect communities by optimising the soft cluster assignment matrix $\mathbf{S}$ using the map equation loss function. Each respective row of $\mathbf{S}$ specifies the fraction of communitiy membership for each respective node. Optimising $\mathbf{S}$ with respect to map equation loss determines whether nodes belong to one or several communities.
>
> > **(4)** _What is the efficiency of Neuromap in the experiments?_
>
> We discuss the complexity in the Complexity paragraph under section 3. Neuromap's time-efficiency depends on the chosen (graph) neural network architecture and the number of chosen maximum number of clusters. As we highlight in the results section, the detected communities' quality also depends on the chosen neural network architecture, which allows users to trade off between time-efficiency and community quality.

---

### Official Review · Reviewer_MNLx · 2023-11-02

**Soundness:** 3 good
**Presentation:** 2 fair
**Contribution:** 2 fair
**Rating:** 3
**Confidence:** 4

**Summary:**

This paper discusses the application of deep learning and graph neural networks (GNNs) to community detection and graph clustering tasks. It highlights the under-explored nature of graph clustering as a primary task for GNNs and the limitations of existing approaches in identifying meaningful clusters. The authors propose a method that bridges the gap between deep learning and network science by optimizing the map equation, an information-theoretic objective function for community detection. The method proposed by the paper is generally novel to me, but the overall way that the paper conveys its idea remains a lot of ambiguity and the results need to be discussed more comprehensively.

**Strengths:**

(1)	The paper tries to use a novel approach to tackle the graph clustering problem, which is significant and has many real-world applications. The paper has addressed the significance of the problem properly. Related works are discussed properly also.
(2)	The paper tries to employ the map equation to solve the conventional graph clustering problem. In this process, the paper makes the optimization process differential to adapt the advanced GNNs to this process. The method is generally novel to me.
(3)	The experiments show that the performance of the proposed model is roughly good.

**Weaknesses:**

(1)	Paragraph 3 of “Introduction”: I don’t think the community detection using GNNs is “under explored”. There are a few works for this task such as [1], [2], [3], [4] and those discussed in the first paragraph of “Related work”.
(2)	Paragraph 1 and 2 of “Background”: I’m still confused about the goal of the map function. For example, what is the “per-step description length”? What is “Huffman code”? I would suggest maybe the author could introduce this in more detail in Appendix.
(3)	Paragraph 3 of “Background”: I would suggest the author to add a figure to illustrate the whole process discussed in the paragraph to make it more readable.
(4)	In “The map equation goes neural”, the paper introduces “S_{n x s}” without introducing s. I would encourage the author to define s the first time they use it.
(5)	In “The map equation goes neural”, I’m still confused about how the model learns S. The paper claims that S is learned via MLP or GNN, but S is a soft cluster assignment matrix.
How could we learn a matrix using MLP or GNN? Is it an output from MLP or GNN? If so, what is the input?
(6)	What is the advantage of the proposed model over traditional ones such as KNN and DeepWalk? The paper discusses the existing approaches in paragraph 2 of “Introduction”, but does not mention the motivation of the proposed one. To me the complexity of KNN is O(nd), where d is the feature dimension, whereas the proposed method has the complexity of O(n^2), which is worse than KNN.
(7)	The results in Table 2 show that DmoN has superior performance than the proposed method in many settings. Why? The paper should discuss this. Also, the proposed method performs badly in “arXiv” dataset, which is also not discussed.
(8)	I would suggest the authors put the caption of the table on the top to make the presentation more formal.
[1] Bruna and Li, Community detection with graph neural networks
[2] Sun et al., Graph neural network encoding for community detection in attribute networks
[3] Luo et al., Detecting communities from heterogeneous graphs: A context path-based graph neural network model
[4] Yuan et al., Community detection with graph neural network using Markov stability

**Questions:**

Please refer to my comments in “Weakness”.

---

> ### Author Response · Authors · 2023-11-21
>
> > This paper discusses the application of deep learning and graph neural networks (GNNs) to community detection and graph clustering tasks. It highlights the under-explored nature of graph clustering as a primary task for GNNs and the limitations of existing approaches in identifying meaningful clusters. The authors propose a method that bridges the gap between deep learning and network science by optimizing the map equation, an information-theoretic objective function for community detection. The method proposed by the paper is generally novel to me, but the overall way that the paper conveys its idea remains a lot of ambiguity and the results need to be discussed more comprehensively.
> >
> > 1. The paper tries to use a novel approach to tackle the graph clustering problem, which is significant and has many real-world applications. The paper has addressed the significance of the problem properly. Related works are discussed properly also.
> > 2. The paper tries to employ the map equation to solve the conventional graph clustering problem. In this process, the paper makes the optimization process differential to adapt the advanced GNNs to this process. The method is generally novel to me.
> > 3. The experiments show that the performance of the proposed model is roughly good.
>
> We thank the reviewer for highlighting the significance and novelty of our approach.
>
> There may be some misunderstandings regarding the scientific problem we consider. We focus on unsupervised community detection based on topological patterns using the information-theoretic objective function known as the map equation. As input, we assume a network with nodes that may or may not be attributed with node features. By expressing the map equation in differentiable tensor form, we enable optimising the map equation with GNNs and gradient descent. However, we neither construct a network from high-dimensional data points, for example, using KNN, nor perform dimensionality reduction, for example, using DeepWalk. Using GNNs to optimise the map equation, we augment topology-based community-detection with node features.
>
> > **(1)** _Paragraph 3 of “Introduction”: I don’t think the community detection using GNNs is “under explored”. There are a few works for this task such as [1], [2], [3], [4] and those discussed in the first paragraph of “Related work”._
> >
> > [1] Bruna and Li, Community detection with graph neural networks \
> > [2] Sun et al., Graph neural network encoding for community detection in attribute networks \
> > [3] Luo et al., Detecting communities from heterogeneous graphs: A context path-based graph neural network model \
> > [4] Yuan et al., Community detection with graph neural network using Markov stability
>
> Thank you for pointing out further related works. We have changed the wording in our manuscript.
>
> > **(2)** _Paragraph 1 and 2 of “Background”: I’m still confused about the goal of the map function. For example, what is the “per-step description length”? What is “Huffman code”? I would suggest maybe the author could introduce this in more detail in Appendix. 3._
> >
> > **(3)** _Paragraph 3 of “Background”: I would suggest the author to add a figure to illustrate the whole process discussed in the paragraph to make it more readable._
>
> We have revised the manuscript and explain the technical details about the map equation in the appendix as the reviewer suggests. We have also added a figure where we use a coding example to explain how the map equation works.
>
> > **(4)** _In “The map equation goes neural”, the paper introduces “S\_{n x s}” without introducing s. I would encourage the author to define s the first time they use it._
>
> Thank your for pointing out the delayed definition of $s$. We had introduced $s$ further down in the text but have now moved it to where we use the symbol for the first time.
>
> > **(5)** _In “The map equation goes neural”, I’m still confused about how the model learns S. The paper claims that S is learned via MLP or GNN, but S is a soft cluster assignment matrix. How could we learn a matrix using MLP or GNN? Is it an output from MLP or GNN? If so, what is the input?_
>
> The input we provide to the MLP or GNN is the node feature matrix. In cases when no node features are available, we use the identity matrix, that is, a one-hot encoding of node identities. The MLP or GNN transforms the node features and outputs a matrix which we interpret as soft cluster assignments. The specific network architecture determines how the graph's adjacecny matrix is used during learning. _Any_ neural network, MLP, GNN, or others can be used to learn a soft cluster assignment matrix. We then use the map equation loss to calculate the quality of this soft cluster assignment matrix and use back propagation to minimise the loss. This way, the GNN clusters the network using the map equation.

---

> > ### Comment · Reviewer_MNLx · 2023-11-22
> > **Reply to author's response**
> >
> > Thank the author for his response. However, I still have the following concerns and therefore I will maintain my score for now.
> >
> > (1) I cannot see the revised paper so that I'm not sure if my concerns have been addressed in the paper.
> >
> > (2) When answering (5), the author said "Any neural network, MLP, GNN, or others can be used to learn a soft cluster assignment matrix.", but I'm still confused how this could be done.
> >
> > (3) I'm still confused about the motivation of the proposed approach over traditional clustering methods such as K means clustering as the author does not explain the heavy complexity caused by the proposed approach.

---

> ### Author Response · Authors · 2023-11-21
>
> > **(6)** _What is the advantage of the proposed model over traditional ones such as KNN and DeepWalk? The paper discusses the existing approaches in paragraph 2 of “Introduction”, but does not mention the motivation of the proposed one. To me the complexity of KNN is O(nd), where d is the feature dimension, whereas the proposed method has the complexity of O(n^2), which is worse than KNN._
>
> The objectives of the map equation, KNN and DeepWalk are quite different.
>
> KNN is an approach that, given d-dimensional data points, can be used to construct a graph where nodes are connected to their K nearest neighbours. However, KNN does not produce a clustering of the nodes.
>
> DeepWalk simulates a large number of short random walks to learn the "context" of a node to perform dimensionality reduction. To detect clusters, however, DeepWalk needs to be combined with another approach such as k-means which does not automatically determine the optimum number of clusters.
>
> The map equation is an objective function for unsupervised community detection. The standard formulation of the map equation seeks to identify groups of nodes that are more densely connected to each other than to the rest. Using the map equation as a loss function for (graph) neural networks enables combining topological features, that is link patterns, with node features for community detection. Importantly, the map equation does not have any hyperparameters. Specifically, it does not require choosing the number of clusters; it is derived from the data using the minimum description length principle.
>
> > **(7)** _The results in Table 2 show that DmoN has superior performance than the proposed method in many settings. Why? The paper should discuss this. Also, the proposed method performs badly in “arXiv” dataset, which is also not discussed._
>
> We have added additional experimental results where Neuromap performs more competitively on the `ogbn-arxiv` dataset. We have also highlighted the best results in Table 2 more clearly. The experiments run by Tsitsulin et al. involve a peculiar choice of the maximum number of clusters -- 16 -- despite the number of ground-truth clusters in `ogbn-arxiv` being 40. As valuably suggested, we ran DMoN on our overlapping community examples, which evidenced DMoN neither easily finds an optimum number of clusters, nor finds reasonable overlapping communities. When run with a larger number of maximum clusters, 100 and 1000, Neuromap performs competitively on the `ogbn-arxiv` dataset among others, which highlights its ability to automatically infer a meaningful number of clusters and avoid over-partitioning, a significant advantage over existing methods.
>
> > **(8)** _I would suggest the authors put the caption of the table on the top to make the presentation more formal._
>
> We have updated the manuscript and placed the table captions above the tables.

---

> ### Author Response · Authors · 2023-11-23
>
> > Thank the author for his response. However, I still have the following concerns and therefore I will maintain my score for now.
>
> We thank the reviewer for the swift response. Unfortunately, we cannot follow the reviewer's assessment; in particular:
>
> > (1) I cannot see the revised paper so that I'm not sure if my concerns have been addressed in the paper.
>
> We have double-checked and confirmed that the PDF on OpenReview is updated. We also tried accessing it without being logged in and could retrieve the revised version. We kindly ask the reviewer to try accessing the PDF again.
>
> > (2) When answering (5), the author said "Any neural network, MLP, GNN, or others can be used to learn a soft cluster assignment matrix.", but I'm still confused how this could be done.
>
> We're unsure what exactly the confusion is. We explain again the general approach below.
>
> Consider the output of a (G)NN which is a $n \times s$ matrix where $n$ is the number of nodes in the graph and $s$ is the maximum possible number of clusters.
> We interpret this matrix as $\mathbf{S}$, a soft cluster assignment of nodes to clusters, and use our adapted map equation loss to compute its quality.
> To optimise $\mathbf{S}$, we use backpropagation to learn weights based on map equation loss.
>
> As input to the (G)NN, we use the node feature matrix $\mathbf{X}$ if node features are available. If no node features are available, we use the identity matrix $\mathbf{I}$ as node features, $\mathbf{X} = \mathbf{I}$.
> The (G)NN's architecture determines how the graph's adjacency matrix $\mathbf{A}$ is used for learning $\mathbf{S}$ from $\mathbf{X}$.
>
> > (3) I'm still confused about the motivation of the proposed approach over traditional clustering methods such as K means clustering as the author does not explain the heavy complexity caused by the proposed approach.
>
> Indeed, there appears to be a misunderstanding regarding the settings of "traditional clustering" in continuous spaces and graph clustering.
> "Traditional" methods such as k-means assume that distances between data points can be measured in order to determine cluster centers.
> In contrast, graph clustering methods such as the map equation do not assume that data points are embedded in a metric space -- clusters are determined from the graph's link patterns.
>
> Clustering is an NP-hard problem such that solving it exactly has exponential complexity.
> Heuristic methods for solving k-means, such as Lloyd's algorithm, are often considered "linear" in time because they return reasonable results after a small number of iterations and can, therefore, be stopped early.
> However, their worst-case complexity is exponential.
> As we explain in our manuscript, the quadratic worst-case complexity for optimising the map equation with GNNs originates from dense matrix-matrix multiplications when setting the maximum possible number of clusters $s$ equal to the number of nodes $n$.
> We have, therefore, recommended in our manuscript to keep $s \ll n$.
> In summary, it seems unfair to compare our quadratic worst-case complexity to the best-case complexity of k-means which not only is due to early stopping, but also applies to different data structures.
>
> Moreover, we emphasise once again that the map equation automatically determines the best number of clusters in a data-driven fashion while "traditional" methods require choosing the number of clusters.

---

### Author Response · Authors · 2023-11-21

Dear Reviewers,

First and foremost, we would like to extend our sincere gratitude for the time and effort you have dedicated to reviewing our manuscript. Your insights and feedback are invaluable to us and the broader academic community.

Upon careful consideration of your comments, we have noted a few instances where there might be some misunderstandings about specific aspects of our work. We appreciate this opportunity to clarify these points to ensure a mutual understanding of our research's intentions and findings.

We understand that the complexity of the subject matter can sometimes lead to different interpretations, and we are grateful for the opportunity to provide additional clarity. We have revised our manuscript to make these points clearer, avoiding any potential confusion for future readers.

We appreciate your contributions to refining our work and hope that our clarifications resonate with your understanding of the subject.
Thank you once again for your valuable feedback and for aiding in the advancement of our research.

Sincerely,\
The Authors

---

### Meta-Review · Area_Chair_vkqz · 2023-12-13

**Metareview:**

This paper is concerned with the application of deep learning and Graph Neural Networks (GNNs) to community detection and graph clustering. It proposes a method that bridges the gap between deep learning and network science by optimizing the map equation, an information-theoretic objective function for community detection. The method proposed in the paper seems sensible; however, the presentation of the paper should be carefully improved, and the experimental results should be made more convincing.

**Justification For Why Not Higher Score:**

The presentation of the paper should be carefully improved, and the experimental results should be made more convincing.

**Justification For Why Not Lower Score:**

The method proposed in the paper seems sensible.

---

### Decision · Program_Chairs · 2024-01-16

Reject